# Skill deficits among foreign-educated immigrants: Evidence from the U.S. PIAAC

**Jason Richwine** *

Center for Immigration Studies, Washington, DC, United States of America

* jmr@cis.org

## Abstract

Researchers have long observed that foreign-educated immigrants earn lower wages and hold less-skilled jobs than U.S. natives who have the same level of educational attainment, but the reasons for the disparity have been less clear. This paper tests the hypothesis favored by the human capital model of earnings and employment–namely, that foreign-educated immigrants struggle in the U.S. labor market primarily because they possess fewer marketable skills than workers with U.S. degrees. Standardized tests administered as part of the Program for the International Assessment of Adult Competencies reveal that foreign-educated immigrants score 0.82 and 0.54 standard deviations lower on measures of literacy and numeracy, respectively, compared to natives who have the same age and educational attainment. The gaps remain significant after controlling for self-assessed English reading ability. When these skill measures are incorporated into regression analyses, the wage and skilled-employment penalties experienced by foreign-educated immigrants fall by half or more, providing strong evidence for the human capital model. However, this analysis cannot rule out additional explanatory factors, such as legal and social obstacles that foreign-educated immigrants may face.

## Introduction

The value of foreign educational credentials in the United States is an ongoing concern of public administrators, policymakers, and researchers. Because most foreign schools operate outside the jurisdiction of U.S. accreditation bodies, the Office of Personnel Management requires an extensive review by a "credential evaluation service" before training from such schools can be used as a qualification for federal employment [1]. Meanwhile, developing more efficient processes to evaluate foreign schooling has become an international priority [2].

In Congress, policymakers looking to attract high-skill immigrants are divided on the value of foreign schooling. The immigration reform bill debated in 2013 would have created a points system that rewards education without distinction between U.S. and "equivalent" foreign degrees [3]. However, under the proposed RAISE Act [4], applicants with professional or doctoral degrees in a STEM field would receive three extra points (10 percent of the total needed to apply) when their degrees are from U.S. institutions.

**Data Availability Statement:** The data underlying the results presented in the study are available from the U.S. National Center for Education Statistics at the URL below. Anyone may apply for and receive a copy. However, due to government

privacy rules, I do not have the right to post the
data publicly myself. https://nces.ed.gov/surveys/
piaac/datafiles.asp.

**Funding:** The author received no specific funding
for this work.

**Competing interests:** The authors have declared
that no competing interests exist.

Researchers seeking to understand why some highly-educated immigrants struggle in the
U.S. labor market have identified foreign credentials as an obstacle to advancement [5–7]. The
issue will grow more salient as the average education level of immigrants rises. According to
the Annual Social and Economic Supplement of the Current Population Survey, 34 percent of
recent working-age immigrants had at least a college degree in 2007, and an identical 34 per-
cent had less than a high school diploma. Even without a deliberate shift toward a high-skill
immigration policy, the share of recent working-age immigrants with at least a bachelor's
degree rose to 49 percent by 2021, while the share without a high school diploma declined to
16 percent. To what extent this improvement in educational attainment reflects an increase in
skills depends in large part on the value of foreign schooling.

As both low-skill and high-skill immigration increased in the U.S. in the late 1990s, Schoeni
[8] demonstrated with Census data that an additional year of schooling among foreign-edu-
cated immigrants increased earnings by a smaller amount than an additional year of schooling
among U.S.-educated immigrants. More recently, Batalova et al. [5] showed that higher levels
of both unemployment and "underemployment"—holding a job that normally requires less
than one's level of educational attainment–could be found among foreign-educated immi-
grants compared to U.S-educated immigrants.

Because their data contain no explicit information on the place of education, the two papers
cited above impute foreign education based on years of schooling and age at arrival. For exam-
ple, in the case of Batalova et al. [5], a college graduate who arrived in the U.S. after the age of
25 is assumed to be foreign-educated. However, immigrants have an incentive to continue
their schooling beyond normal completion ages [9], as even small amounts of U.S. education
can increase their overall returns to education [10]. The measurement error caused by these
imputations likely understates the difference in value between foreign and U.S. schooling.

To identify foreign education more precisely, Fogg and Harrington [6, 7] used the National
Survey of College Graduates (NSCG), which includes the country where each respondent's col-
lege degree was acquired. Compared to U.S-educated immigrants in the 2003 NSCG, foreign-
educated immigrants from all regions except the UK and Canada were found to have lower
earnings and higher rates of underemployment. The deficits compared to U.S.-educated col-
lege graduates were especially pronounced among immigrants educated in Latin America,
with wages 38 percent lower and a rate of underemployment that was more than 2.5 times
greater.

The lower return to foreign education is not unique to the U.S. Using data from Israel,
Friedberg [11] showed that the return to Israeli schooling for foreign-born residents was
higher than the return to foreign schooling. Belmonte et al. [12] found that college-educated
residents of the EU were more likely to be unemployed or underemployed when they have a
foreign degree from a non-EU country. The lower value of foreign schooling has also been
demonstrated in Canada [13] and Australia [14], where having a non-English-speaking back-
ground exacerbates the penalty for foreign education.

Foreign-educated immigrants have less success in their adopted countries' labor markets
for several possible reasons. They may be unfamiliar with regulations, networking, and licens-
ing requirements [15], while others may arrive without authorization [16] or on temporary
visas that restrict their job options [17]. In addition, employers may unfairly dismiss foreign
credentials as inadequate [18]. However, the focus of the present paper is on the hypothesis
favored by the human capital model of earnings and employment–that is, foreign-educated
immigrants possess fewer marketable skills than people educated in their host countries.

Assessing the human-capital theory has been surprisingly difficult. Most datasets allow for
only indirect skill measures, such as self-assessed English ability, quality of the home country
educational system [19], or college major. By contrast, the literacy and numeracy tests

administered by the Program for the International Assessment of Adult Competencies (PIAAC) are designed to be direct and robust measures of skill, and scores on the PIAAC tests have been found to be significant determinants of earnings [20].

When Ferrer et al. [21] used a predecessor of the PIAAC literacy test in a study of immigrants in Ontario, controlling for literacy skills eliminated about two thirds of the wage gap between foreign and Canadian schooling among college-educated immigrants. However, Lancee and Bol [22] showed that controlling for PIAAC skills eliminated at most one third of the wage gap between Western and non-Western-educated immigrants in 11 Western European countries, leaving a large role for credentialing in the European labor market.

The contribution of this paper is to directly compare the skills of U.S. natives, U.S.-educated immigrants, and foreign-educated immigrants in the U.S. using PIAAC data, and then explore the degree to which skill differences can explain differences in labor market outcomes. It finds that the concerns about foreign schooling are well founded. Compared to natives with the same age and educational attainment, foreign-educated immigrants score 0.82 and 0.54 standard deviations lower on tests of literacy and numeracy, respectively. After further adjusting for self-assessed English reading ability, the gaps remain significant at 0.41 and 0.18 standard deviations. Controlling for test scores reduces the wage and skilled-employment penalties experienced by foreign-degree holders by half or more, providing strong support for the human capital model in the U.S. However, small sample sizes preclude a more detailed analysis that would be necessary to rule out additional causes of labor market differences.

## Data and methods

### PIAAC

Coordinated internationally by the OECD to facilitate cross-country comparisons of adult skills, the PIAAC was administered by the National Center for Education Statistics (NCES) in the U.S. The NCES conducted three waves of testing that it combined to yield a nationally representative sample of about 12,000 Americans, ages 16 to 74, who were tested between 2012 and 2017 [23]. Because of the level of detail available in the PIAAC data set, NCES required a license (#21040007) to access the data and has subsequently provided written confirmation that this study does not disclose any personally identifiable information. A license to access the restricted PIAAC data can be obtained by any qualified researcher who submits an application to NCES [24].

PIAAC interviews began with a background questionnaire and were followed by three computer-adaptive tests covering literacy, numeracy, and computer-based problem solving. The tests themselves were strictly in English, leading to interpretive issues that will be discussed more below. However, the PIAAC offered several accommodations to encourage participation, including a Spanish translation of the background questionnaire, a paper-and-pencil alternative for participants who are uncomfortable with a computer, and a more basic test of reading for those who have a low level of literacy. Only about 2 percent of the initial respondents had to be excluded from the test because of a language or cognitive handicap that prevented them from communicating their basic demographic information [25].

### Key terms

Conceptually, *literacy* is not merely the ability to read sentences. PIAAC's definition of literacy is "understanding, evaluating, using, and engaging with written text to participate in society, to achieve one's goals, and to develop one's knowledge and potential" [25]. By this definition, even native English speakers receive low scores on the test when their literacy skills are not well developed.

PIAAC defines *numeracy* as "the ability to access, use, interpret, and communicate mathematical information and ideas, [and] to engage in and manage mathematical demands of a range of situations in adult life" [25]. The PIAAC does not cover advanced math such as algebra or calculus. Instead, test items are based on challenges that might be encountered in ordinary life, from basic arithmetic to drawing inferences from a chart. Although these tasks may seem straightforward, scores on the numeracy test vary widely even among educated natives.

What PIAAC calls *problem-solving in technology-rich environments* (PST) is based at least as much on adeptness with computers as it is on problem-solving ability. It involves "using digital technology, communication tools, and networks to acquire and evaluate information, communicate with others, and perform practical tasks" [25]. Put more concretely, test items require participants to perform computer-based tasks similar to what an executive assistant might do in an office.

In this study, an immigrant is someone who was born outside of the U.S. (Due to the lack of citizenship data in the PIAAC, this definition of an immigrant departs slightly from the traditional Census Bureau definition, which is any U.S. resident who was not a citizen at birth.) Foreign-degree holders are defined as immigrants who arrived in the U.S. either in the same year or after the year in which they completed their highest level of education. As noted above, some earlier research imputed foreign-degree status based solely on age at arrival, but the PIAAC allows for more precise identification by including the age at which immigrants completed their education. Note that an immigrant's status as a foreign-degree holder depends only on the highest degree. If an immigrant receives a bachelor's degree in a foreign nation but then receives a master's degree in the U.S., that immigrant counts as U.S.-educated.

While foreign-educated immigrants nearly always arrive in the U.S. as adults, about two thirds of U.S.-educated immigrants came to the U.S. when they were still minors. This "1.5 generation"–situated between the first generation that arrived in adulthood and the second generation that was born in the U.S.—traditionally outperforms the first generation in the labor market [26], and the advantage of U.S. education is likely a key reason why. Because the sample sizes are small, and the primary focus here is foreign degrees, this study does not separate the "1.5" generation from other U.S.-educated immigrants. However, S1 Table shows that skill differences between foreign- and U.S.-educated immigrants are not the result of the latter simply spending more time in the U.S.

## Literacy and numeracy test scores

The number and distribution of PIAAC test-takers by foreign-degree status and educational attainment are provided in Table 1. To comply with disclosure restrictions set by the NCES, all unweighted sample sizes have been rounded to the nearest 10. The "missing" row in Table 1 includes not only the respondents whose education data was originally missing, but also the roughly 70 natives and 50 immigrants whose education was recoded to missing because their ages of completion appear to be too young for the degrees they claim. (In this paper, the minimum ages for a high school diploma, some college, and a bachelor's or advanced degree are set at 16, 18, and 20, respectively. Excluding these discrepant cases from all education-based analyses shrinks the observed gap in test scores between natives and foreign-educated immigrants).

Table 1 shows that immigrants with U.S. degrees generally have higher levels of educational attainment than natives, while foreign-educated immigrants have levels of attainment that are considerably lower.

To reduce the burden of the survey, PIAAC respondents take different portions of the tests and are subsequently assigned ten "plausible values" to represent their true scores [27]. Both the "piaactools" [28] and "repest" [29] programs for STATA were used to calculate group means and standard errors from these plausible values.

**Table 1. Educational distribution of test-takers with valid PIAAC scores.**

| Highest Schooling | Native | | Immigrant (U.S. Degree) | | Immigrant (Foreign Degree) | |
|---|---|---|---|---|---|---|
| | n | col. % | n | col. % | n | col. % |
| Advanced | 980 | 10.1% | 160 | 23.7% | 70 | 9.2% |
| Bachelor's | 1,750 | 17.6% | 140 | 17.0% | 120 | 14.6% |
| Some College | 1,740 | 18.5% | 110 | 17.1% | 50 | 8.3% |
| High School | 4,410 | 41.3% | 260 | 32.6% | 230 | 32.5% |
| Less Than High School | 1,500 | 11.8% | 70 | 8.9% | 190 | 28.8% |
| (Missing) | 80 | 0.7% | 10 | 0.6% | 40 | 6.7% |
| Total | 10,470 | 100% | 750 | 100% | 700 | 100% |

To comply with government disclosure restrictions, sample sizes have been rounded to the nearest 10. Individual rows may not sum to the "Total" row due to rounding.

n = sample size, col. % = weighted proportion of the column at a given education level

A common problem with average test scores is that they have little meaning without reference to the full score distribution. To make group comparisons interpretable, the average literacy and numeracy scores of different groups are first presented in bar charts as percentiles on the overall U.S. distribution.

Subsequent regression analyses express literacy and numeracy score differences in standard deviations (SDs). These analyses allow scores to be adjusted for two baseline characteristics in addition to education. The first characteristic is age. For reasons possibly related to test familiarity, younger adults tend to score higher on the PIAAC [30]. The second characteristic is self-assessed English reading ability. The ability to read English is an important threshold skill measured by the PIAAC tests. However, PIAAC test scores would add little analytic value if they reflected no more than immigrants' level of familiarity with English. Adjusting for self-assessed English reading ability helps to focus score differences on the broader skills that the tests were intended to measure. Table 2 displays the distribution of self-assessed English reading ability by immigration status.

## PST test scores

All PIAAC tests were administered on a computer, unless respondents had no experience with computers, failed a basic test of computer skills (such as how to use a mouse), or simply refused to use a computer. The non-computer respondents took paper-and-pencil versions of the literacy and numeracy tests. However, because computer competency is intrinsic to the

**Table 2. Distribution of self-assessed English reading ability among test-takers with valid PIAAC scores.**

| Reads English... | Natives | | Immigrant (U.S. Degree) | | Immigrant (Foreign Degree) | |
|---|---|---|---|---|---|---|
| | n | col. % | n | col. % | n | col. % |
| ...Very Well | 9,570 | 91.1% | 540 | 71.1% | 240 | 32.8% |
| ...Well | 730 | 7.3% | 180 | 24.4% | 210 | 31.5% |
| ...Not Well | 120 | 1.2% | 30 | 3.7% | 170 | 23.9% |
| ...Not At All | 40 | 0.4% | 10 | 0.7% | 70 | 11.8% |
| Total | 10,470 | 100% | 750 | 100% | 700 | 100% |

To comply with government disclosure restrictions, sample sizes have been rounded to the nearest 10. Individual rows may not sum to the "Total" row due to rounding.

n = unweighted sample size; col. % = weighted proportion of the column with a given English ability

PST test, there was no alternative version of the PST test offered. As a result, 18.5 percent of PST scores are missing.

The missing cases present a methodological challenge. A complete-case analysis would cause the PST score averages to be upward biased, since the excluded respondents would surely score at the lower end of the distribution. Alternatively, scores could be imputed based on the known scores of similar respondents. The imputation procedure would be difficult, however, because the likelihood of having a missing score depends inherently on the skill measured by the test itself.

The solution here is to analyze PST score *levels* designated by PIAAC rather than scale scores. Each respondent's scale score is converted to one of four PST levels based on thresholds established by PIAAC. Level 0 requires short and straightforward tasks, while tasks at higher levels become increasingly complicated, with Level 3 requiring a high degree of unguided problem solving. Importantly, respondents with missing scores are assumed to score at Level 0. This assumption is a reasonable compromise that sacrifices some precision in order to avoid the bias of dropped cases or imputation based on questionable models. Simple percentages are first used to compare average PST levels across groups, followed by an ordered logit regression that allows for tests of significance and additional controls.

## Wage and employment regressions

After presenting group differences in test scores, the analysis moves on to establishing the wage penalty experienced by foreign-degree holders and exploring how much of it might be explained by differences in skills. Using the framework established by Mincer [31], the log of hourly wages is regressed on different sets of worker characteristics, including separate dummy variables for U.S.-educated immigrants and foreign-educated immigrants.

In the baseline specification, only age, educational attainment, and self-assessed English reading ability are included as controls, but then PIAAC skill measures are added to assess the change in the wage penalty for foreign degrees. The stepwise regressions eventually include a full set of standard covariates, including actual experience, the square of experience, tenure at the current job, race, gender, marital status, and region. A final regression goes beyond the human capital model by including a set of controls for each worker's occupational skill *requirements*, as defined by the International Standard Classification of Occupations (ISCO).

All regressions are limited to full-time employees (not self-employed individuals) between the ages of 18 and 64. Workers report their earnings and work hours as part of the background questionnaire, and then PIAAC converts their earnings to the hourly wage rates used in this study. Finally, workers with unreasonably high or low hourly wages (roughly the top and bottom 1 percent) are excluded from the wage regressions. This exclusion improves the overall model fit but does not change the estimated coefficients in a meaningful way.

In addition to experiencing a wage penalty, immigrants with foreign degrees are also likely to be "underemployed," meaning holding a job that normally requires less than their level of educational attainment. As discussed by Flisi et al. [32], there are several ways to identify workers in the PIAAC data who are underemployed. One is to define a worker as underemployed when the worker's own education exceeds the average or modal level of education within his or her occupation. Unfortunately, the sample size of the U.S. PIAAC is too small to determine the distribution of education in many of the less common occupations. An alternative is to rely on each PIAAC worker's individual assessment of the education level that is required for his or her job. The drawback of this approach, however, is that the standard for determining the required level is subjective and inconsistent across respondents.

For a simple and consistent measure that is available for nearly all workers in the PIAAC, this analysis uses the four levels of ISCO-defined occupational skill requirements–*elementary*,

*semi-skilled blue collar*, *semi-skilled white collar*, and *skilled*. Those levels constitute the dependent variable in an ordered logit that controls for age, education, and self-assessed English reading ability in the baseline model. The coefficients on the U.S.-educated and foreign-educated immigrant variables indicate whether immigrants are less likely to hold skilled jobs compared to natives with similar characteristics. As with wages, PIAAC test scores and other explanatory variables are gradually added to the regression.

The PIAAC data are robust enough to test the hypothesis that skill differences are associated with a substantial part of the wage and skilled-employment penalties experienced by foreign-educated immigrants. However, regressions limited to smaller demographic groups are not feasible. The small sample size, combined with PIAAC's use of "plausible values" that stand in for each respondent's test scores, are major sources of imprecision. Therefore, detailed subgroup analyses will have to wait for additional waves of PIAAC testing.

## Results

### Test score differentials

Fig 1 compares the percentile scores of natives, U.S.-educated immigrants, and foreign-educated immigrants on the PIAAC literacy test. It is visually evident that U.S.-educated immigrants score slightly lower than natives, while foreign-educated immigrants score far lower. The pattern holds even within major educational groups. Among PIAAC respondents with bachelor's degrees, for example, natives score at the 73rd percentile of the U.S. population, and U.S.-educated immigrants reach the 62nd percentile. However, foreign-educated immigrants with bachelor's degrees score at just the 38th percentile, which is lower than the score achieved by natives with only a high school diploma.

Table 3 offers a more formal comparison, with all score differences standardized and adjusted for age differences. Column I indicates that U.S.-educated immigrants score 0.21 SDs lower than natives, while foreign-educated immigrants score 1.06 SDs lower. When education controls are added in Column II, the gap between natives and U.S.-educated immigrants actually increases to 0.35 SDs due to the latter group having higher levels of education. By contrast, the gap between natives and foreign-educated immigrants decreases to 0.82 SDs. Within specific educational levels (Columns III through VII), gaps between natives and immigrants remain large and significant.

The bottom half of Table 3 repeats the analysis from the top half but adds a control for self-assessed English reading ability. Immigrants score substantially better after this adjustment, with the overall gap between natives and foreign-educated immigrants reduced from 1.06 SDs to 0.36 SDs. Nevertheless, most of the remaining differences in the foreign-educated row are still large and significant, which suggests that the observed test-score gap reflects more than just inexperience with English. Interestingly, the reductions in the immigrant-native gaps after controlling for English ability appear to be smaller among the well-educated respondents in Columns III and IV compared to the less-educated respondents in Columns V through VII.

Fig 2 and Table 4 repeat the above analysis with the PIAAC numeracy test. The native-immigrant gaps are smaller here. In fact, among advanced degree holders, natives and U.S.-educated immigrants have roughly the same numeracy scores. Nevertheless, the same patterns that emerged in the literacy data are also evident with numeracy. Foreign-educated immigrants trail both natives and U.S.-educated immigrants by wide margins, including within educational categories, and the differences substantially shrink but do not generally disappear when controlling for self-assessed English reading ability. The numeracy gaps in Column IV are especially notable given immigrants' greater tendency to study STEM fields in college.

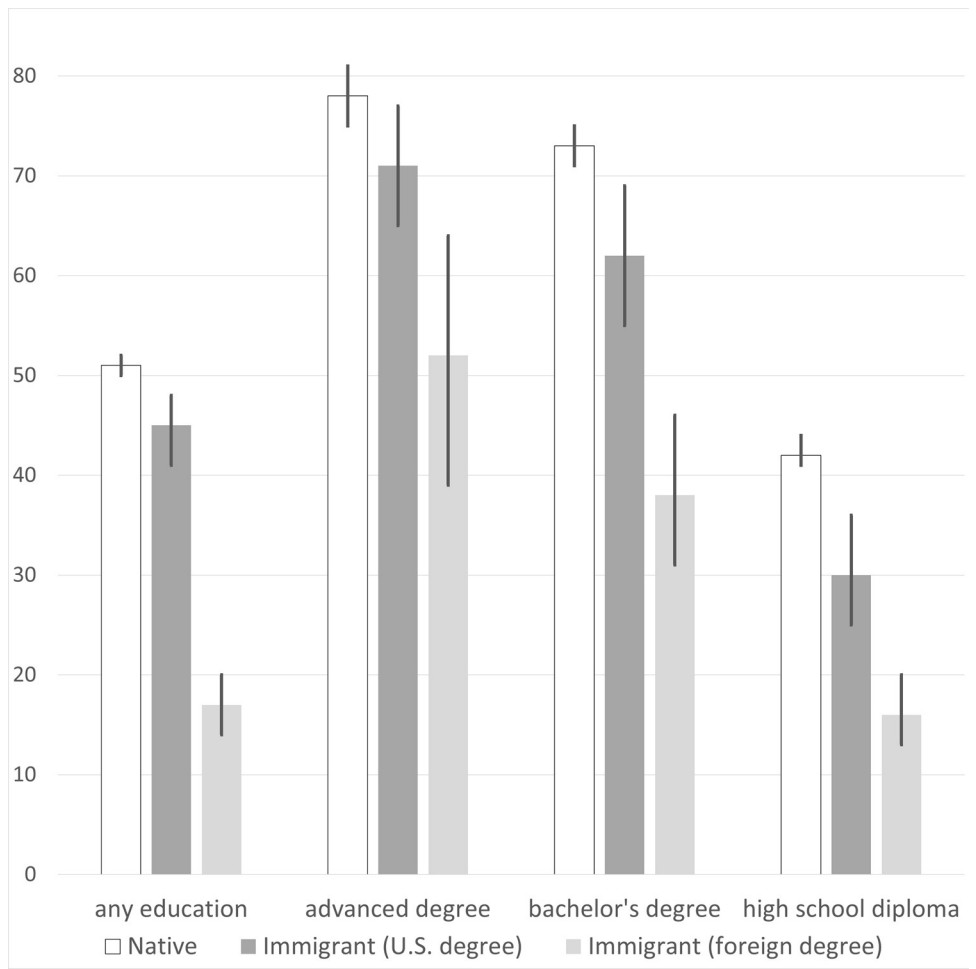

**Fig 1. Average percentile scores on PIAAC literacy, by immigrant-education group.** The vertical line at the top of each bar indicates the 95% confidence interval. The scores are not age-adjusted.

(Among working-age college graduates in the 2019 American Community Survey, 30 percent of immigrants have STEM degrees, compared to 18 percent of natives).

As explained in the methods section, scores on the computer-focused PST are grouped into four levels and analyzed as a categorical variable. Fig 3 shows that 76 percent of foreign-educated immigrants score at the lowest level (Level 0), compared to 40 percent of U.S.-educated immigrants and 34 percent of natives. Figs 4–6 display the level breakdowns within various education groups. Compared to foreign-educated immigrants, natives and U.S.-educated immigrants are generally skewed further toward the higher levels.

To compare PST levels more rigorously, Table 5 displays the results of an ordered logit regression that adjusts test scores for age. Each cell contains an odds ratio and an associated standard error. Column I indicates that foreign-educated immigrants have an odds of scoring at a higher level that is just 0.15 times as large as the odds that natives have. "Scoring at a higher level" refers to any group of levels that are higher than a reference group of levels–e.g., scoring at levels 1 through 3 versus 0, scoring at levels 2 or 3 versus 0 or 1, and so on. Odds ratios less than one indicate that immigrants score at lower levels than natives. As with literacy and numeracy, adjusting for self-assessed English ability substantially shrinks but does not eliminate the performance gaps between natives and foreign-educated immigrants.

**Table 3. Immigrant-native differences in PIAAC literacy scores.**

|  | I | II | III | IV | V | VI | VII |
|---|---|---|---|---|---|---|---|
|  | **Full Sample, No Education Controls** | **Full Sample, Education Controls** | **Only Advanced Degrees** | **Only Bachelor's Degrees** | **Only Some College** | **Only High School Diplomas** | **Only Less Than High School** |
| Without English Ability Control |  |  |  |  |  |  |  |
| U.S.-educated Immigrants | -0.21** | -0.35** | -0.34** | -0.44** | -0.51** | -0.43** | -0.37* |
|  | (0.05) | (0.04) | (0.10) | (0.11) | (0.12) | (0.09) | (0.16) |
| Foreign-educated Immigrants | -1.06** | -0.82** | -0.92** | -1.11** | -0.85** | -0.87** | -0.75** |
|  | (0.07) | (0.06) | (0.21) | (0.12) | (0.23) | (0.09) | (0.10) |
| With English Ability Control |  |  |  |  |  |  |  |
| U.S.-educated Immigrants | -0.05 | -0.24** | -0.27** | -0.36** | -0.38** | -0.26** | -0.17 |
|  | (0.05) | (0.04) | (0.10) | (0.11) | (0.12) | (0.09) | (0.16) |
| Foreign-educated Immigrants | -0.36** | -0.41** | -0.69** | -0.91** | -0.52* | -0.41** | -0.14 |
|  | (0.06) | (0.06) | (0.19) | (0.14) | (0.24) | (0.09) | (0.11) |

The table displays the standardized differences in test scores between immigrants and natives with varying adjustments for educational attainment and self-assessed English reading ability. All differences are age-adjusted. Negative numbers imply a native advantage. Standard errors are in parentheses.

* $p < 0.05$,

** $p < 0.01$

## Effect of controlling for test scores on wage and employment disparities

Table 6 displays the results of nested regressions that examine how controlling for PIAAC test scores affects the wage penalty experienced by foreign-educated immigrants. The baseline regression in Column I establishes that foreign-educated immigrants who have the same age, educational attainment, and self-assessed English reading ability as natives nevertheless receive wages that are $1 - \exp(-0.12) = 11$ percent lower. Is this penalty due to residual skill differences not captured by the baseline control variables, as the human capital model would predict, or is it due to other factors?

To start answering that question, the model in Column II adds the literacy, numeracy, and PST test scores as control variables on top of the baseline model. Doing so cuts the foreign-degree wage penalty in half, from 11 percent in Column I to a statistically insignificant 5 percent in Column II. The last row of Table 5 shows that the change in the foreign-educated coefficient between columns is highly significant.

Column III adds experience and demographic characteristics to the regression, and Column IV adds a set of dummies for the ISCO skill level of each worker's occupation. After dropping significantly between Column I and Column II, the coefficient on foreign-educated immigrants declines only marginally with the addition of these other covariates. The standard errors are too high to conclude that controlling for test scores eliminates the *entire* wage gap observed at baseline between natives and foreign-educated immigrants. However, test scores are clearly associated with a substantial portion of the gap.

Table 7 demonstrates that controlling for test scores also helps to increase the likelihood that foreign-educated immigrants will hold skilled jobs. In the baseline regression, immigrants with foreign schooling have an odds of holding a job with a higher skill requirement that is just 70 percent as large as the odds for natives. The odds ratio significantly increases to 90 percent in Column II with the inclusion of test scores. Adding the full set of demographic controls in Column III causes no significant change in the foreign-educated coefficient. Therefore, the tendency for foreign-educated immigrants to hold lower-skilled jobs compared to natives appears to be driven in large part by differences in human capital. As with wage differences,

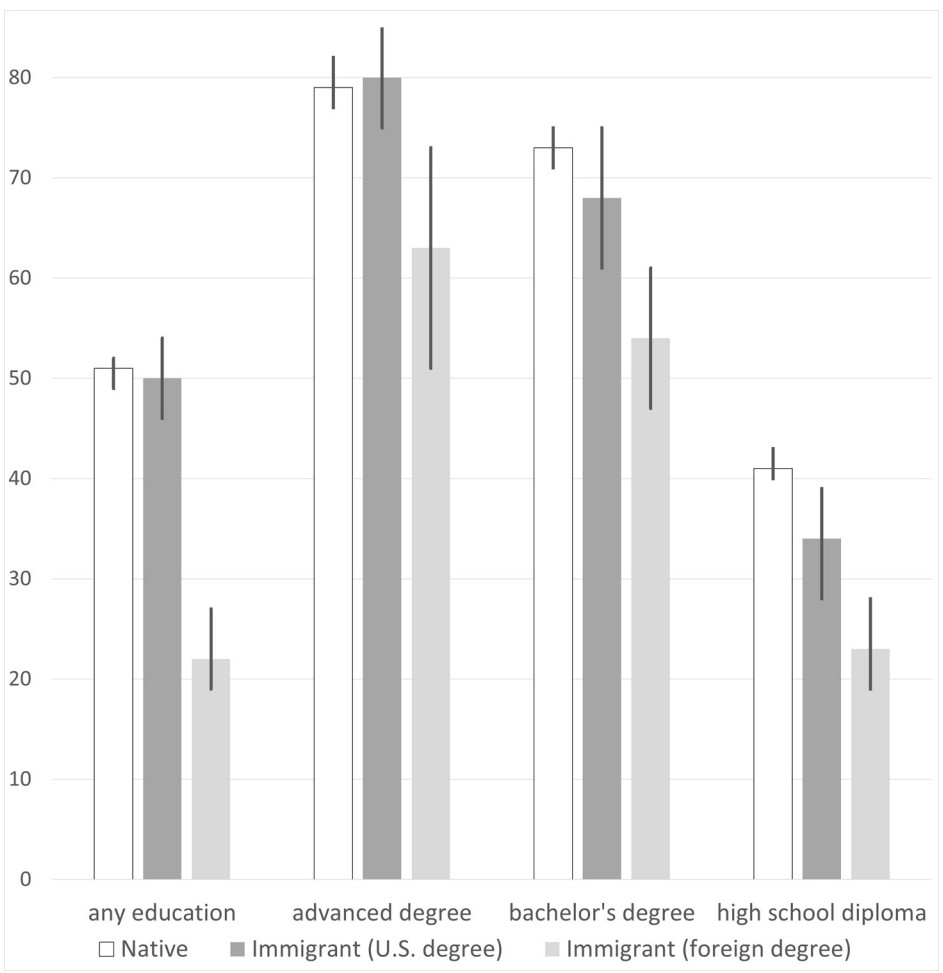

**Fig 2. Average percentile scores on PIAAC numeracy, by immigrant-education group.** The vertical line at the top of each bar indicates the 95% confidence interval. The scores are not age-adjusted.

however, the analysis cannot completely rule out factors beyond human capital that may be influencing the disparity.

## Discussion

This study has shown that large and significant skill gaps exist between immigrants and natives who have the same age and educational attainment. However, the size of those skill gaps depends on where immigrants were educated. After controlling for age and educational attainment, U.S.-educated immigrants score 0.35 SDs lower than natives on the PIAAC literacy test, while foreign-educated immigrants score 0.82 SDs lower. Controlling for self-assessed English reading ability reduces the native advantage over U.S.-educated and foreign-educated immigrants to 0.24 SDs and 0.41 SDs, respectively, but these gaps remain significant. The same pattern of score differences appears on the PIAAC numeracy and computer-focused PST tests.

The divergence between U.S.- and foreign-educated immigrants becomes even more evident when comparing wage and employment outcomes. Despite the lower test scores, U.S.-educated immigrants earn about the same wages and take on skilled jobs at least as often as natives with comparable baseline characteristics of age, education, and English reading ability.

**Table 4. Immigrant-native differences in PIAAC numeracy scores.**

| | I | II | III | IV | V | VI | VII |
|---|---|---|---|---|---|---|---|
| | Full Sample, No Education Controls | Full Sample, Education Controls | Only Advanced Degrees | Only Bachelor's Degrees | Only Some College | Only High School Diplomas | Only Less Than High School |
| Without English Ability Control | | | | | | | |
| U.S.-educated Immigrants | -0.06 | -0.21** | -0.03 | -0.26* | -0.41** | -0.28** | -0.33* |
| | (0.05) | (0.04) | (0.11) | (0.12) | (0.12) | (0.09) | (0.14) |
| Foreign-educated Immigrants | -0.81** | -0.54** | -0.62** | -0.63** | -0.43 | -0.55** | -0.66** |
| | (0.07) | (0.06) | (0.19) | (0.10) | (0.25) | (0.09) | (0.11) |
| With English Ability Control | | | | | | | |
| U.S.-educated Immigrants | 0.09 | -0.11** | 0.04 | -0.21 | -0.31* | -0.14 | -0.14 |
| | (0.05) | (0.04) | (0.11) | (0.13) | (0.12) | (0.08) | (0.13) |
| Foreign-educated Immigrants | -0.13* | -0.18** | -0.42* | -0.48** | -0.19 | -0.16 | -0.04 |
| | (0.06) | (0.05) | (0.17) | (0.12) | (0.26) | (0.08) | (0.11) |

The table displays the standardized differences in test scores between immigrants and natives with varying adjustments for educational attainment and self-assessed English reading ability. Negative numbers imply a native advantage. Standard errors are in parentheses.

* $p < 0.05$,

** $p < 0.01$

After controlling for test scores, U.S.-educated immigrants may even outperform natives in the labor market, depending on the model. By contrast, foreign-educated immigrants earn 11 percent less than natives at baseline and have an odds of moving into higher-skill jobs that is only 70 percent as large. Depending on the regression specification, roughly half or more of the magnitudes of these gaps can be attributed to a skill deficit, providing support for the human capital model's theoretical explanation of wage and employment differences.

The results suggest that the rising level of educational attainment among recent immigrants to the U.S. should be viewed with some skepticism. Compared to people with the same educational attainment in the U.S., foreign-educated immigrants possess fewer skills that are

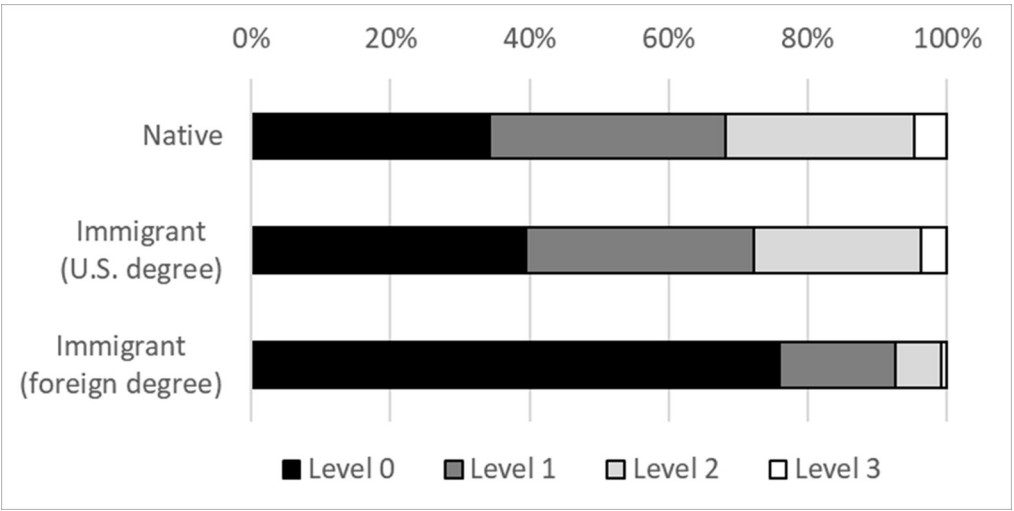

**Fig 3. Distribution of PIAAC PSL category scores, education = any.** The figure displays the percentage scoring at four different levels of the PST test. Higher levels require more sophisticated problem solving. The percentages are not age-adjusted.

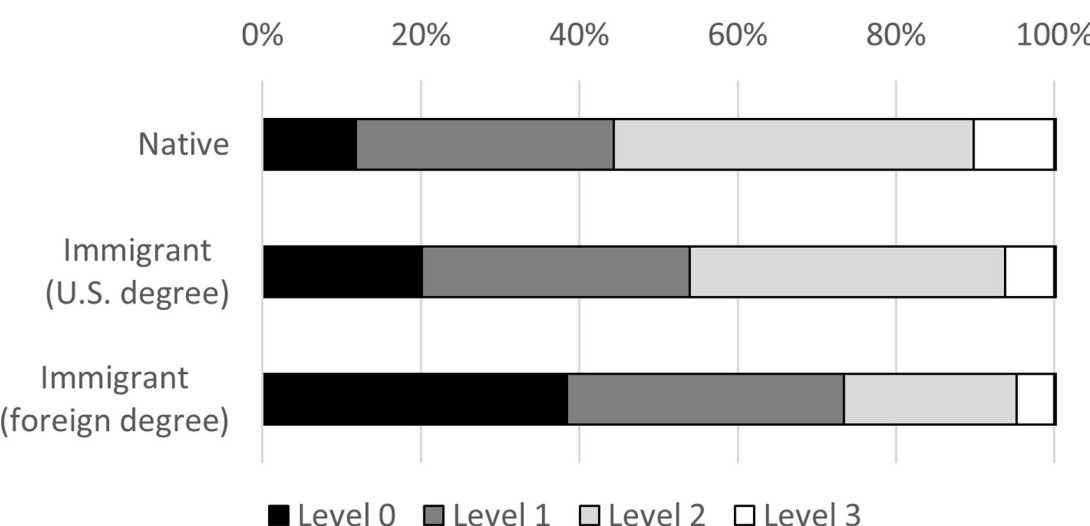

**Fig 4. Distribution of PIAAC PSL category scores, education = advanced.** The figure displays the percentage scoring at four different levels of the PST test. Higher levels require more sophisticated problem solving. The percentages are not age-adjusted.

rewarded in the U.S. labor market. Therefore, it is a mistake to assume that the U.S. has been moving toward a de facto "high-skill" immigration system. If Congress does seek an explicit policy change that would attract more high-skill immigrants, it should consider giving greater weight to U.S.-based degrees or incorporating strict rules for foreign accreditation. Congress may also want to consider using skill tests as an alternative or supplement to educational attainment.

Although the analysis above shows that skill deficits are likely the most important reason that foreign-educated immigrants struggle in the U.S. labor market, the regression estimates lack sufficient precision to rule out additional explanatory factors, such as insufficient net-working, unfamiliarity with licensing requirements, and legal restrictions on employment. All of these structural factors are plausibly correlated with having a foreign degree. Indeed, S2 Table suggests in a supplemental analysis that the wage and skilled-employment gaps that

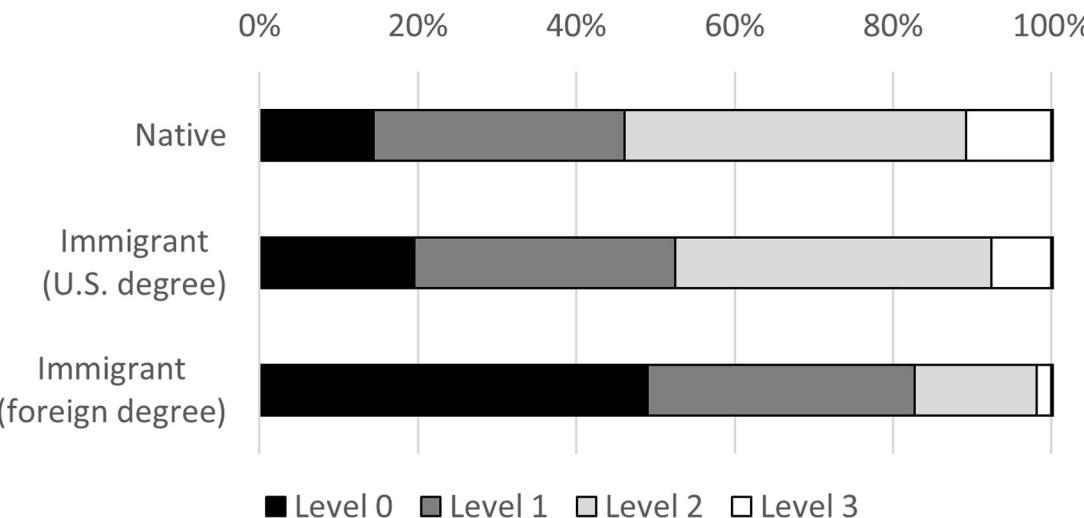

**Fig 5. Distribution of PIAAC PSL category scores, education = bachelor's.** The figure displays the percentage scoring at four different levels of the PST test. Higher levels require more sophisticated problem solving. The percentages are not age-adjusted.

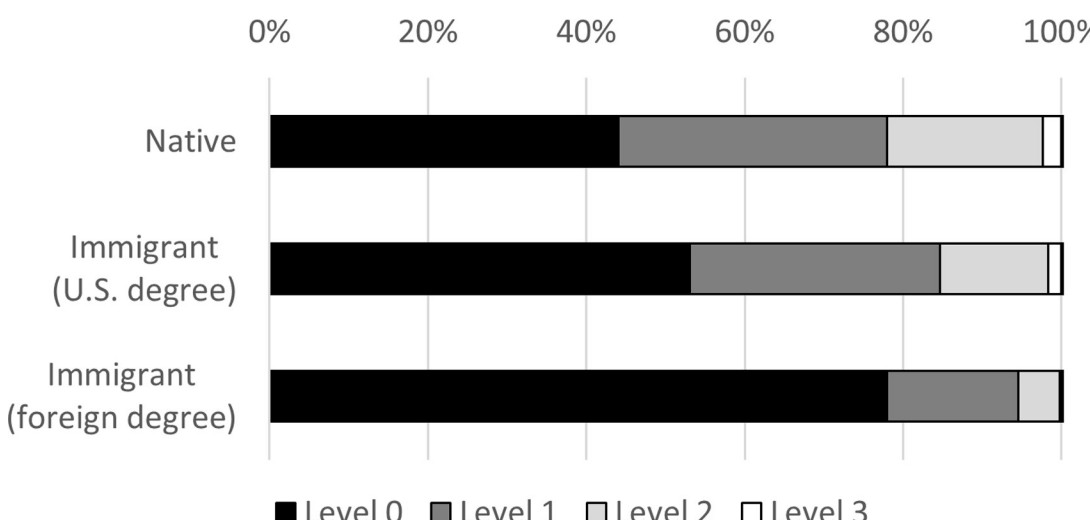

**Fig 6. Distribution of PIAAC PSL category scores, education = high school.** The figure displays the percentage scoring at four different levels of the PST test. Higher levels require more sophisticated problem solving. The percentages are not age-adjusted.

remain after controlling for PIAAC scores are substantial for recent arrivals but fade to approximately zero over time. This result is consistent with a model in which structural factors initially play some role in labor market disparities, but skill differences gradually become the dominant factor. However, more research with a larger sample is needed to confirm this tentative result.

The advantages of this study over previous analyses of foreign degrees in the U.S. are twofold. First, it uses direct measures of skills in the form of standardized tests, while some previous studies either use indirect proxies for skills or have no skill measures at all beyond educational attainment. Second, this study is able to more accurately distinguish between U.S. and foreign education by comparing the age of arrival with the age of degree completion–the latter of which

**Table 5. Immigrant-native differences in PIAAC PST scores.**

| | I | II | III | IV | V | VI | VII |
|---|---|---|---|---|---|---|---|
| | **Full Sample, No Education Controls** | **Full Sample, Education Controls** | **Only Advanced Degrees** | **Only Bachelor's Degrees** | **Only Some College** | **Only High School Diplomas** | **Only Less Than High School** |
| **Without English Ability Control** | | | | | | | |
| U.S.-educated Immigrants | 0.67** | 0.48** | 0.47** | 0.52** | 0.37** | 0.50** | 0.49** |
| | (0.07) | (0.05) | (0.12) | (0.13) | (0.10) | (0.08) | (0.20) |
| Foreign-educated Immigrants | 0.15** | 0.17** | 0.21** | 0.15** | 0.16** | 0.21** | 0.11** |
| | (0.02) | (0.03) | (0.07) | (0.04) | (0.07) | (0.04) | (0.08) |
| **With English Ability Control** | | | | | | | |
| U.S.-educated Immigrants | 0.86 | 0.57** | 0.53** | 0.61** | 0.47** | 0.62** | 0.70 |
| | (0.09) | (0.07) | (0.13) | (0.15) | (0.13) | (0.10) | (0.31) |
| Foreign-educated Immigrants | 0.40** | 0.29** | 0.28** | 0.21** | 0.27** | 0.39** | 0.40 |
| | (0.06) | (0.05) | (0.09) | (0.06) | (0.11) | (0.09) | (0.35) |

The table displays odds ratios of scoring at a higher level on the four-level PST test, with varying adjustments for educational attainment and self-assessed English reading ability. Odds ratios less than one imply a native advantage. All odds ratios are age-adjusted. Standard errors are in parentheses.

* $p < 0.05$,

** $p < 0.01$

**Table 6. Effect of controlling for test scores on the immigrant-native wage gap.**

| | I | II | III | IV |
|---|---|---|---|---|
| | **Baseline** | **Plus Test Scores** | **Plus Full Demos** | **Plus Occ** |
| Immigrants (ref. = Natives) | | | | |
| U.S.-educated | 0.04 | 0.07 | 0.05 | 0.04 |
| | (0.04) | (0.04) | (0.04) | (0.03) |
| Foreign-educated | -0.12** | -0.05 | -0.04 | -0.03 |
| | (0.04) | (0.04) | (0.05) | (0.04) |
| Age (ref. = 18–24) | | | | |
| 25–34 | 0.29** | 0.27** | 0.14** | 0.12** |
| | (0.02) | (0.02) | (0.02) | (0.02) |
| 35–44 | 0.49** | 0.49** | 0.25** | 0.23** |
| | (0.03) | (0.03) | (0.04) | (0.04) |
| 45–54 | 0.51** | 0.53** | 0.21** | 0.21** |
| | (0.03) | (0.02) | (0.04) | (0.04) |
| 55–64 | 0.59** | 0.63** | 0.27** | 0.27** |
| | (0.03) | (0.03) | (0.05) | (0.04) |
| Education (ref. = no HS) | | | | |
| High School (HS) | 0.20** | 0.12** | 0.10** | 0.08** |
| | (0.03) | (0.03) | (0.03) | (0.03) |
| Some College | 0.36** | 0.23** | 0.22** | 0.15** |
| | (0.03) | (0.04) | (0.03) | (0.03) |
| Bachelor's | 0.66** | 0.44** | 0.45** | 0.31** |
| | (0.04) | (0.04) | (0.03) | (0.03) |
| Advanced | 0.81** | 0.57** | 0.58** | 0.41** |
| | (0.05) | (0.06) | (0.05) | (0.05) |
| Read English (ref. = Very Well) | | | | |
| Well | -0.03 | 0.02 | -0.02 | -0.01 |
| | (0.03) | (0.03) | (0.03) | (0.03) |
| Not Well | -0.20** | -0.11* | -0.17** | -0.14** |
| | (0.05) | (0.05) | (0.05) | (0.05) |
| Not at All | -0.32** | -0.21* | -0.21* | -0.15 |
| | (0.08) | (0.09) | (0.09) | (0.08) |
| Test Scores | | | | |
| Literacy | | -0.003 | 0.036 | 0.031 |
| | | (0.034) | (0.032) | (0.03) |
| Numeracy | | 0.126** | 0.057 | 0.056* |
| | | (0.029) | (0.030) | (0.03) |
| PST (ref. = Level 0) | | | | |
| Level 1 | | 0.04 | 0.07* | 0.03 |
| | | (0.03) | (0.03) | (0.03) |
| Level 2 | | 0.10* | 0.12** | 0.07* |
| | | (0.04) | (0.04) | (0.04) |
| Level 3 | | 0.13 | 0.16* | 0.10 |
| | | (0.07) | (0.07) | (0.06) |
| Experience | | | 0.0143** | 0.0127** |
| | | | (0.0037) | (0.0033) |
| Experience$^2$ | | | -0.0002** | -0.0002** |
| | | | (0.0001) | (0.0001) |

(*Continued*)

**Table 6.** (Continued)

| | I | II | III | IV |
|---|---|---|---|---|
| | **Baseline** | **Plus Test Scores** | **Plus Full Demos** | **Plus Occ** |
| Tenure | | | 0.0099** | 0.0094** |
| | | | (0.0012) | (0.0012) |
| Woman | | | -0.17** | -0.19** |
| | | | (0.02) | (0.02) |
| Partnered | | | 0.09** | 0.07** |
| | | | (0.02) | (0.02) |
| Race (ref. = White) | | | | |
| Black | | | 0.0003 | 0.0171 |
| | | | (0.03) | (0.03) |
| Hispanic | | | -0.0003 | 0.0009 |
| | | | (0.04) | (0.04) |
| Other | | | 0.07 | 0.06 |
| | | | (0.04) | (0.03) |
| Region (ref. = Northeast) | | | | |
| Midwest | | | -0.07 | -0.06 |
| | | | (0.04) | (0.04) |
| South | | | -0.11* | -0.11** |
| | | | (0.04) | (0.04) |
| West | | | 0.02 | 0.01 |
| | | | (0.04) | (0.04) |
| Occupation (ref. = Elementary) | | | | |
| Semi-skilled blue collar | | | | 0.15** |
| | | | | (0.04) |
| Semi-skilled white collar | | | | 0.07* |
| | | | | (0.04) |
| Skilled | | | | 0.38** |
| | | | | (0.03) |
| Constant | 2.19** | 1.69** | 1.81** | 1.77** |
| | (0.04) | (0.09) | (0.09) | (0.09) |
| $r^2$ | 0.35 | 0.39 | 0.46 | 0.50 |
| N | 4,160 | 4,160 | 4,160 | 4,160 |
| Δ Foreign-Educated: Current Col. minus Prev. Col. | | 0.064** | 0.012 | 0.014 |
| | | (0.013) | (0.020) | (0.012) |

The table displays the coefficients from an OLS regression of the log of hourly wages on different sets of worker characteristics. Standard errors are in parentheses.

* $p < 0.05$,

** $p < 0.01$. To comply with government disclosure restrictions, sample sizes have been rounded to the nearest 10.

is not available in most Census data. Given the strong evidence afforded by this data in favor of the human capital model, future research must not assume that foreign-educated immigrants in general have less success in the labor market solely because of structural hurdles.

The principal limitation of this study, however, is the small sample size afforded by the U.S. PIAAC. While previous studies have been able to conduct detailed subgroup analyses, including evaluating degrees from individual source countries or regions, the present data set is not large enough to examine small demographic groups. It is unrealistic to expect that any nationally representative data set that includes standardized test scores could ever be as large as the

**Table 7. Effect of controlling for test scores on the immigrant-native gap in occupational skill requirements.**

| | I | II | III |
|---|---|---|---|
| | **Baseline** | **Plus Test Scores** | **Plus Full Demos** |
| Immigrants (ref. = Natives) | | | |
| U.S.-educated | 1.45 | 1.70* | 1.46 |
| | (0.23) | (0.27) | (0.26) |
| Foreign-educated | 0.70* | 0.90 | 0.85 |
| | (0.15) | (0.19) | (0.17) |
| Age (ref. = 18–24) | | | |
| 25–34 | 1.63* | 1.65* | 1.30 |
| | (0.26) | (0.29) | (0.25) |
| 35–44 | 1.85** | 2.01** | 1.27 |
| | (0.30) | (0.34) | (0.28) |
| 45–54 | 1.65* | 1.96** | 1.02 |
| | (0.27) | (0.34) | (0.23) |
| 55–64 | 1.96** | 2.65** | 1.22 |
| | (0.32) | (0.46) | (0.34) |
| Education (ref. = no HS) | | | |
| High School (HS) | 1.80** | 1.50* | 1.51* |
| | (0.28) | (0.24) | (0.24) |
| Some College | 4.17** | 3.06** | 3.05** |
| | (0.82) | (0.65) | (0.64) |
| Bachelor's | 17.80** | 11.53** | 11.84** |
| | (3.12) | (2.35) | (2.43) |
| Advanced | 68.04** | 43.98** | 43.77** |
| | (19.63) | (13.73) | (13.75) |
| Read English (ref. = Very Well) | | | |
| Well | 0.62** | 0.77 | 0.86 |
| | (0.10) | (0.13) | (0.15) |
| Not Well | 0.46** | 0.58* | 0.58* |
| | (0.14) | (0.18) | (0.19) |
| Not at All | 0.19** | 0.20** | 0.22** |
| | (0.08) | (0.09) | (0.10) |
| Test Scores | | | |
| Literacy | | 1.35* | 1.13 |
| | | (0.16) | (0.13) |
| Numeracy | | 0.74** | 0.93 |
| | | (0.08) | (0.11) |
| PST (ref. = Level 0) | | | |
| Level 1 | | 1.93** | 1.86** |
| | | (0.31) | (0.29) |
| Level 2 | | 2.75** | 2.72** |
| | | (0.63) | (0.59) |
| Level 3 | | 3.95 | 4.21 |
| | | (1.57) | (1.73) |
| Experience | | | 1.035* |
| | | | (0.0155) |
| Experience$^2$ | | | 1.000 |
| | | | (0.0003) |

(*Continued*)

**Table 7.** (Continued)

|  | I | II | III |
|---|---|---|---|
|  | **Baseline** | **Plus Test Scores** | **Plus Full Demos** |
| Tenure |  |  | 1.011* |
|  |  |  | (0.0050) |
| Woman |  |  | 2.44** |
|  |  |  | (0.25) |
| Partnered |  |  | 1.25* |
|  |  |  | (0.11) |
| Race (ref. = White) |  |  |  |
| Black |  |  | 0.93 |
|  |  |  | (0.11) |
| Hispanic |  |  | 1.10 |
|  |  |  | (0.16) |
| Other |  |  | 1.14 |
|  |  |  | (0.24) |
| Region (ref. = Northeast) |  |  |  |
| Midwest |  |  | 0.81 |
|  |  |  | (0.10) |
| South |  |  | 1.02 |
|  |  |  | (0.12) |
| West |  |  | 1.24 |
|  |  |  | (0.20) |
| N | 4,160 | 4,160 | 4,160 |
| Δ Foreign-Educated (current col. / prev. col.) |  | 1.28** | 0.95 |
|  |  | (0.06) | (0.08) |

The table displays the odds ratios from an ordered logit regression of occupational skill requirements on different sets of worker characteristics. The skill requirements are grouped into four categories: *elementary*, *semi-skilled blue-collar*, *semi-skilled white collar*, and *skilled*. Standard errors are in parentheses.

* $p < 0.05$,

** $p < 0.01$. To comply with government disclosure restrictions, sample sizes have been rounded to the nearest 10.

regular Census products such as the Current Population Survey. Nonetheless, future waves of PIAAC data collection would expand the range of potential research, including a more in-depth analysis of how the length of residence in the U.S. mediates the relationship between labor market disparities and measured skills, as well as an examination of how the skills of newly-arriving foreign-educated immigrants are changing over time.

# Supporting information

**S1 Table. Effect of years of residence on test-score differentials between foreign—and U.S-educated immigrants.** (a) Standardized differences in literacy or numeracy test scores between foreign- and U.S.-educated immigrants, with and without an adjustment for years of U.S. residence. Negative numbers imply a U.S.-educated advantage. (b) Foreign-educated odds ratios of scoring at a higher level on the four-level PST test relative to U.S.-educated immigrants, with and without an adjustment for years of U.S. residence. Odds ratios less than one imply a U.S.-educated advantage. Residence is a categorical variable with ten different values. All comparisons are adjusted for age and self-assessed English reading ability. Standard

errors are in parentheses. $^{*} p < 0.05$, $^{**} p < 0.01$.
(DOCX)

**S2 Table. Effect of controlling for test scores on the immigrant-native wage and skilled employment gaps, with immigrants separated by years of U.S. residency.** The first two columns display the coefficients from an OLS regression of the log of hourly wages on different sets of worker characteristics. The differences between the foreign-educated coefficients in the first and second columns are displayed in the bottom section. The last two columns display the odds ratios from ordered logit regressions of occupational skill requirements on different sets of worker characteristics. The ratios of the foreign-educated coefficients in the third and fourth columns are displayed in the bottom section. Standard errors are in parentheses. $^{*} p < 0.05$, $^{**} p < 0.01$. To comply with government disclosure restrictions, sample sizes have been rounded to the nearest 10.
(DOCX)

## Author Contributions

**Conceptualization:** Jason Richwine.

**Formal analysis:** Jason Richwine.

**Investigation:** Jason Richwine.

**Methodology:** Jason Richwine.

**Project administration:** Jason Richwine.

**Writing – original draft:** Jason Richwine.

**Writing – review & editing:** Jason Richwine.

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
