## [Decision Letter · Decision Letter 0]

1 Jul 2022

PONE-D-22-15058Skill deficits among foreign-educated immigrants: Evidence from the U.S. PIAACPLOS ONE

Dear Dr. Richwine,

Thank you for submitting your manuscript to PLOS ONE. After careful consideration, we feel that it has merit but does not fully meet PLOS ONE’s publication criteria as it currently stands. Therefore, we invite you to submit a revised version of the manuscript that addresses the points raised during the review process.

The paper investigates an important aspect, backed by theoretical assumptions of human capital, of foreign-educated immigrants in the U.S. labour market. The two reviewers agree that the research and its results can contribute to existing literature on the topic, and they are positive about accepting it for publication. As academic editor, I wish to draw your attention to the many remarks they made and ask you to revise your manuscript accordingly. Please, spend some more thoughtful time on responding to the reviewers' comments and send us your corrected and revised text. 

We look forward to receiving your revised manuscript.

Kind regards,

István Tarrósy, PhD

Academic Editor

PLOS ONE

Journal Requirements:

Reviewers' comments:

Reviewer's Responses to Questions

**Comments to the Author**

1. Is the manuscript technically sound, and do the data support the conclusions?

Reviewer #1: Yes

Reviewer #2: Yes

2. Has the statistical analysis been performed appropriately and rigorously? 

Reviewer #1: Yes

Reviewer #2: Yes

3. Have the authors made all data underlying the findings in their manuscript fully available?

Reviewer #1: Yes

Reviewer #2: Yes

4. Is the manuscript presented in an intelligible fashion and written in standard English?

Reviewer #1: Yes

Reviewer #2: Yes

5. Review Comments to the Author

Reviewer #1: My uploaded review contains several proposal in time being and regional details. This recent publication makes evidences about the importance of human resources US level, but the next step is integration. Migration decision is a chain where you are making a more or less blind step, if it is successful than win-win. But if not, both the side will be frustrated, looking for other option. So I am sure we do make effort to deal with the real circumstances the given geographical places. It is named subsidiarity of mobility.

These are my comments:

Before reviewing the manuscript topic, my 40 years researcher's assumption was that a foreign degree compares **different institutional accreditations on countries level** and that, like all migrants, they have to do in host country their professional work under different institutional and professional systemic conditions. Like engineer, doctors, accountant… The literature summarizes this transit terms of **re-engineering**. It means we can conclude any hypothesis by time series transversal (to follow those subpopulations by calendar years or even more) data analysis. It is a pioneer report to deal with this topic and necessary, but encourage us for follow up.

The experience of Central Europe shows that foreigners can later **get a job that matches their qualifications on a kind of ladder.** It means their early wages might be as lower as their domestic experienced colleagues. So the impact for labor market is not direct. The nominal number is not quality.   

Here, there is a clash between **individual temporal** gain, loss, and the **country/region gains** in the host labor market. In manuscript is not reach and mentioned the regional level, although in the spot are the process fit together. This is the **subsidiarity of migration**. The gap among the individual and nations  takes time and effort to be understand each goals.  The higher the degree, the more everything has to be met and it takes time. It explains the lower wages, the higher acceptance of recruitment. Or region, city can add some /accommodation, allowances more conveniences, advantages to transit situation. The empirical study required in this direction.

The foreign schooling has become an international priority, meanwhile the **evaluation based **not on the certification even more the reference of person, famous institute or team building.  Some times the institution profile shows the cost of degree, probability to be successful in graduation. See from developing country student choose a country to reach European certification on a **cheap cost**.

**Definition** plays an important issue: like an immigrant is someone who was born outside of the U.S. What about the English-speaking countries? More precise identification by including the age at which immigrants completed their education. It is high preference, an easy way to enter US on student F1 visa, and then to follow your path.

The fig3. confirm by higher skill/education comparing the native population immigrants have chance, for US has advantage to recruit better educated people on their own cost. The successful migrant feels their cost and effort return. It is direct surplus to US, but in migration strategy which firstly based on security, and after that the principle of human benefit.

References are punction and accessible, these are the most important citations. 

The topic is important reference for discussion about point system. The article is **highly contributed to raise new questions**. To show a detailed direction of research interest or empirical survey.

The topic is not only by researcher noticed but the employee to moderate the labor market temporary redundancies, to recruit for a special activity skilled worker. In this case you can found extra wages. It is a rising amount to recruit human resources by agencies or representatives of TNCs.

How can we develop a reliable system to measure the competencies, human development, to see the time being this question? The practical aspect of the topic has both sided interest the motivation of migrant and the host countries or regional authorities.

The statistical methods are clear, the explanation is ready.   You can refer to it in the text, how from skill deficit to have transient to surplus. To take next step to be more positive.

Reviewer #2: Are differences in earnings and employment between immigrant and native-born attributed to varying skill, i.e. numeracy, literacy and computer competencies. The objective is clear and analysis well executed. The author finds, consistent with human capital theory, that lower scores on standardized tests of numeracy, literacy, and computer-based tasks are important sources of immigrant lower earnings and employment outcomes. This paper makes insightful contributions to immigrant integration and employment scholarships and has important policy implications.

1) Data and methods – I would like more information on your DV and controls. Are earnings self-reported? Do respondents report hourly earnings? Do you transform annual earnings into hourly earnings using weeks and hours worked? Does your sample include self-employed individuals? I suspect not, because the DV is “earnings”, but this should be clear. Essentially, I would like a little more information on your variables. Methodologically, this was a clear and well-executed analysis.

2) Robustness checks. The decision to drop high and low earners was fine, but I would like a robustness check to confirm that the findings do not change substantially without this.

3) Analytic sample size. The sample changes across models. Is this because of rounding requirements or does the underlying sample change between models? If so, please justify.

4) What about the 1.5 generation? Do you control for them? Why or why not? Are they predominantly in the immigrant – US degree category? Are your findings re: 1.5 gen consistent with prior analysis, and the human capital model you present?

5) Imprecise / precise etc. – In more than one place you refer to your analysis and findings as imprecise / less precise. I assume you are referring to unobserved heterogeneity and small n (some clarification on page 24, line 405), but it might be worthwhile to clarify earlier for the non-specialist or qualitative audience.

6) Page 4, line 82 – add a few sources to support the claim that employers may unfairly dismiss foreign credentials as inadequate. Perhaps, Oreopoulos (2011)?

7) Page 5, line 89 – Are these standard or novel measures of literacy, numeracy, and PST skill? In other words, have others used this data for similar purposes?

8) Page 5, line 104 – tell the reader why you were unable to conduct a “detailed analysis”, I suspect it’s a sample size issue, but make that clear.

9) Page 21, the interpretation of the results from table 7 are light. Consider adding a sentence or two to aid the reader in understanding the importance of these findings.

Reference

Oreopoulos, Philip. 2011. “Why Do Skilled Immigrants Struggle in the Labor Market? A Field Experiment with Thirteen Thousand Resumes.” American Economic Journal: Economic Policy 3(4):148–71.

6. PLOS authors have the option to publish the peer review history of their article (what does this mean?). If published, this will include your full peer review and any attached files.

Reviewer #1: No

Reviewer #2: No

---

## [Author Response · Author response to Decision Letter 0]

30 Jul 2022

I attached a document called "Response to Reviewers." Here is the full text of that document:

I thank both reviewers for their very helpful suggestions, all of which I have incorporated into the revised manuscript.

Response to Reviewer #1

The first reviewer’s point that the difference in foreign degrees could be due to different accreditation standards and practices across countries, and that such issues would perhaps decline in importance as immigrants acculturate, is well taken. I revised the manuscript to address this issue in three ways:

First, I added to the Discussion section the role of time in potentially reducing the legal and social obstacles to advancement for foreign-educated immigrants. 

Second, I conducted a new regression analysis that splits the foreign-educated immigrants into two groups of roughly equal size – those who have been in the U.S. for less than 15 years, and those who have been in the U.S. for at least 15 years. This analysis shows that the wage and skilled-employment gaps that remain after controlling for PIAAC scores are substantial for recent arrivals but fade to approximately zero over time. This result is consistent with a model in which structural factors initially play some role in labor market disparities, but skill differences gradually become the dominant factor. I considered putting this new regression analysis in the main text, but ultimately I decided to include it as a supplemental table (Table S2) that I reference in the Discussion section. The reason is that the analysis requires splitting an already small sample of foreign-educated immigrants in half, and I want to be careful about not attempting to draw strong conclusions from regression results that have high standard errors.

Third, I re-emphasized in the concluding paragraph that further research with larger datasets can hopefully address the reviewer’s note that country- and region-based variation are important in establishing the value of foreign degrees.

Response to Reviewer #2

1) I have added text to the regression portion of the methods section that clarifies earnings are self-reported and converted by PIAAC into an hourly wage based on responses to other questions about work time. I have also added that earnings do not include self-employment.

2) I have added text to the same portion of the methods section indicating that there are no meaningful changes to coefficients when including very high and low earnings, but the model fit (as measured by r-squared) does go down.

3) Thanks for pointing this out. The sample size had been changing due primarily to some missing occupational skill scores. I have re-run the wage and employment regressions with a consistent sample and updated Tables 6 and 7 accordingly. The results are little changed, but I agree it is better to keep the sample sizes the same across models.

4) Good thought. I have added a paragraph to the Key Terms section acknowledging how common 1.5 gen immigrants are in the U.S.-educated group and citing researching indicating that they typically outperform the 1.0 generation. I explain that it’s not practical to separate 1.5 from 1.0 in this study; however, I have added a table in the Supplemental Information section showing that test scores differences between foreign- and U.S.-educated immigrants are not a function of different amounts of time spent in the U.S.

5) I have added or modified text in several places to be clearer about what I mean by “imprecision,” including at the end of the Introduction.

6) Thanks. I have incorporated Oreopoulos into the discussion, and in the same section I’ve added citations to a few other papers that discuss potential legal and social obstacles to immigrant advancement.

7) They’re standard measures of skill that have been used in wage regressions in the past. I’ve modified the text on that page to make this clearer, and I’ve rewritten the description of the cited paper to say explicitly that the PIAAC scores have been found to be significant determinants of earnings.

8) I rewrote the end of that paragraph to explain in more detail why sample size limitations preclude detailed analyses of subgroups.

9) I added and modified text in that paragraph to explain that skilled employment results are similar to the wage results in that they show test scores can explain much of the disparity, but additional factors cannot be ruled out.

---

## [Decision Letter · Decision Letter 1]

18 Aug 2022

Skill deficits among foreign-educated immigrants: Evidence from the U.S. PIAAC

PONE-D-22-15058R1

Dear Dr. Richwine,

We’re pleased to inform you that your manuscript has been judged scientifically suitable for publication and will be formally accepted for publication once it meets all outstanding technical requirements.

Kind regards,

István Tarrósy, PhD

Academic Editor

PLOS ONE

Additional Editor Comments (optional):

The author has dealt with all the comments of the reviewers thoroughly, and the article has been improved with this round of revision. I am now happy to support its acceptance for publication.

Reviewers' comments:

Reviewer's Responses to Questions

**Comments to the Author**

1. If the authors have adequately addressed your comments raised in a previous round of review and you feel that this manuscript is now acceptable for publication, you may indicate that here to bypass the “Comments to the Author” section, enter your conflict of interest statement in the “Confidential to Editor” section, and submit your "Accept" recommendation.

Reviewer #1: (No Response)

Reviewer #2: All comments have been addressed

2. Is the manuscript technically sound, and do the data support the conclusions?

Reviewer #1: Yes

Reviewer #2: Yes

3. Has the statistical analysis been performed appropriately and rigorously? 

Reviewer #1: I Don't Know

Reviewer #2: Yes

4. Have the authors made all data underlying the findings in their manuscript fully available?

Reviewer #1: Yes

Reviewer #2: Yes

5. Is the manuscript presented in an intelligible fashion and written in standard English?

Reviewer #1: Yes

Reviewer #2: Yes

6. Review Comments to the Author

Reviewer #1: The topic of skill has a raising long-term impact. We have learned from the history of flow, those who are skilled has more welcome in the host society and easier to integrate in new conditions. From the point of host countries, the recruitment is stronger who can fill the domestic labor market. So, regional professional specific management is important. The past shows the skill, qualification is essential, but the personal habits, competency is the key issue of any success. The paper has an important study on this area, but from the other hand not follow the immigrants in time being. The really selection on labor market is how is he able to re-engineering his skill. Some professions are essential in accreditation of company activity, like doctor, scientist or lawyer. I encourage the author in the future widen his research how university teaching for host and how are migrants are able to widen their ability to the local chance. This paper gives to readers a shot but not a flow.

Reviewer #2: The author(s) have addressed all my questions and concerns. I am pleased with the thoroughness of the revision.

7. PLOS authors have the option to publish the peer review history of their article (what does this mean?). If published, this will include your full peer review and any attached files.

Reviewer #1: No

Reviewer #2: No

---

## [Editor Report · Acceptance letter]

22 Aug 2022

PONE-D-22-15058R1 

Skill deficits among foreign-educated immigrants: Evidence from the U.S. PIAAC 

Dear Dr. Richwine:

I'm pleased to inform you that your manuscript has been deemed suitable for publication in PLOS ONE. Congratulations! Your manuscript is now with our production department. 

Kind regards, 

on behalf of

Dr. István Tarrósy 

Academic Editor

PLOS ONE